# Towards *n*-type conductivity in hexagonal boron nitride

Shiqiang Lu [1,3], Peng Shen[1,3], Hongye Zhang[1], Guozhen Liu[1], Bin Guo[1], Yehang Cai[1], Han Chen[1], Feiya Xu[1], Tongchang Zheng[2], Fuchun Xu[1], Xiaohong Chen[1], Duanjun Cai [1 ✉] & Junyong Kang[1]

Asymmetric transport characteristic in *n*- and *p*-type conductivity has long been a fundamental difficulty in wide bandgap semiconductors. Hexagonal boron nitride (h-BN) can achieve *p*-type conduction, however, the *n*-type conductivity still remains unavailable. Here, we demonstrate a concept of orbital split induced level engineering through sacrificial impurity coupling and the realization of efficient *n*-type transport in 2D h-BN monolayer. We find that the O $2p_z$ orbital has both symmetry and energy matching to the Ge $4p_z$ orbital, which promises a strong coupling. The introduction of side-by-side O to Ge donor can effectively push up the donor level by the formation of another sacrificial deep level. We discover that a Ge-$O_2$ trimer brings the extremely shallow donor level and very low ionization energy. By low-pressure chemical vapor deposition method, we obtain the in-situ Ge-O doping in h-BN monolayer and successfully achieve both through-plane (~100 nA) and in-plane (~20 nA) *n*-type conduction. We fabricate a vertically-stacked *n*-hBN/*p*-GaN heterojunction and show distinct rectification characteristics. The sacrificial impurity coupling method provides a highly viable route to overcome the *n*-type limitation of h-BN and paves the way for the future 2D optoelectronic devices.

[1] Fujian Key Laboratory of Semiconductor Materials and Applications, CI Center for OSED, College of Physical Science and Technology, Xiamen University, Xiamen 361005, China. [2] Department of Physics, School of Science, Jimei University, Xiamen 361021, China. [3] These authors contributed equally: Shiqiang Lu, Peng Shen. ✉email: dcai@xmu.edu.cn

As a novel ultra-wide bandgap semiconductor, hexagonal boron nitride (h-BN) has a two-dimensional (2D) layered honeycomb structure and attracted enormous attentions[1]. Due to its extraordinary physical properties such as low dielectric constant, high chemical stability, thermal conductivity, electrical resistivity, and mechanical strength, h-BN has broad applications in 2D electronic devices as a gate dielectric layer or a protecting encapsulator[2–5]. In addition, as a functional semiconductor itself, h-BN shows excellent optical properties. The ultra-wide bandgap (~6.5 eV) of h-BN promises its important role in deep ultraviolet (DUV) optoelectronics[6,7]. Due to the 2D confinement feature, the exciton binding energy of h-BN is as huge as 740 meV, which exhibits a great advantage for radiative emission[8–11]. In 2004, room temperature lasing at 215 nm has already been reported for h-BN via accelerated electron excitation[12] and in 2009, a 225 nm plane-emission device equipped with a field-emission electron excitation source was fabricated[13]. This work strongly demonstrated the great potential of h-BN in developing novel DUV optoelectronic devices. However, the most important p-n junction for highly efficient devices is still unavailable for h-BN mainly because of the absence of n-type conducting layer.

Bipolar conducting semiconductors (p-type and n-type layers) provide the most crucial building blocks for constructing electronic and optoelectronic devices such as p-n junction diodes, bipolar transistors, detectors, light-emitting diodes, and laser diodes[14]. However, wide bandgap semiconductors, e.g., ZnO, AlGaN, Ga$_2$O$_3$, diamond and h-BN, suffer a serious asymmetric problem in n- and p-type carrier concentrations and their transport characteristics[15–18]. This is fundamentally due to the relatively low valence band maximum (VBM) or high conduction band minimum (CBM). Consequently, impurities tend to form deep levels located in the middle of the bandgap, behaving as deep acceptors or donors (Fig. S1a–c)[19,20]. The p-type h-BN has been achieved by either Mg-doping[11,21,22] or producing boron vacancies[23]. It has been found that the VBM of h-BN appears relatively higher than that of AlN by 0.67 eV, which leads to the shallow acceptor level[24,25]. In other words, together with the ultrawide bandgap, the position of CBM in h-BN could be extremely high at the same time (Fig. S2). It has been reported that usual donor impurities including C, Si, O and etc. can only form very deep levels against efficient ionization (>0.6 eV)[26–29]. Consequently, the difficulty in n-type doping becomes extremely hard to overcome by conventional method. Up-to-date, reliable realization of effective n-type conductivity in h-BN has yet to be achieved.

In this work, we proposed a method of orbital split induced level engineering through sacrificial impurity coupling and achieved effective n-type conduction in monolayer h-BN. First-principles calculations were employed to investigate the strong orbital coupling between the donor impurity atom and various sacrificial coordinating atoms. A symmetry and energy matching have been observed in $p_z$ orbitals from Ge and O. Energy level engineering was conducted for pushing the Ge donor level closely towards the conduction band. By using low-pressure chemical vapor deposition system (LPCVD), germanium dioxide (GeO$_2$) was used as an impurity precursor for in situ Ge-O doping in h-BN monolayer. As a result, the efficient n-type conductivity of h-BN monolayer was successfully obtained for the first time and confirmed with simultaneous through-plane and in-plane n-type conducting currents. A diode structure with a vertically stacked n-hBN/p-GaN junction was fabricated showing excellent rectification behavior.

## Results and discussions
### Sacrificial impurity coupling induced orbital level engineering.
Systematic first-principles calculations of potential donor impurities

including C, Si, Ge, Sn and etc. have been performed by doping into the h-BN monolayer system. Figure 1a shows the energy band structure of the substitutional Ge$_B$ doped h-BN system. The Ge$_B$ introduces an impurity level (Level-1) at about 230 meV below CBM and behaves as a deep donor. The activation of Ge$_B$ dopant for providing additional electron carriers is energetically costly. Moreover, as shown in Fig. 1b, the carriers from Level-1 demonstrate a strong localization behavior, which is confined around the core of Ge$_B$ dopant due to the weak coupling of $p_z$ orbital with the neighboring N atoms. These results indicate the typical asymmetric doping characteristic and the n-type difficulty in the ultra-wide bandgap h-BN, which are mainly attributed to the deep donor level with high ionization energy and the localized carrier distribution. To overcome the n-type difficulty in h-BN, how to reduce the ionization energy of the donor is the crucial point. For single impurity doping, the impurity level is usually determined by the interaction between impurity and host atom orbitals. If introducing another foreign impurity as a near neighbor, the impurity level could be modulated by the additional orbital coupling and hybridization between impurities.

According to this idea, we investigated the coupling between Ge and neighboring O dopants. As shown in Fig. S3a–d and Fig. 1c–e, the band structures of Ge impurity coupled with one O, two O, and three O atoms were calculated. One can observe that for the Ge-O dimer, there are two separated impurity levels within the bandgap (Fig. 1c). The upper (Level-1') and lower (Level-2) impurity levels can be recognized as the antibonding and bonding states upon the Ge-O orbital hybridization. Compared to the Ge$_B$ level (Level-1) in Fig. 1a, the Level-1' undergoes a downward shift to a deeper energy position, about 480 meV below CBM. It is interesting to see that when higher coordination was applied with Ge-O$_2$ trimer and Ge-O$_3$ tetramer (Fig. 1d–e), the coupling strength between Ge and O is significantly enhanced and the Level-1' could be effectively pushed upward, closer to CBM. The ionization energy of Level-1' for the Ge-O$_2$ and Ge-O$_3$ cases are 89 and 0 meV, respectively, much shallower than that of Level-1. Meanwhile, Level-2 moves downward by 1.4 and 2.4 eV, respectively. These results reveal that the lower Level-2 goes deeper into the middle of bandgap as a sacrifice and meanwhile pushes up the Level-1' towards CBM. The extremely shallow donor Level-1' can then efficiently provide additional electrons for the n-type conductivity of h-BN monolayer.

A detailed investigation into the orbital constituents of these impurity levels in the partial density of states (DOSs) plot (Fig. 1f) further reveals that the upper shallow donor level is originated from the hybridization between Ge 4 $p_z$ and O 2 $p_z$ orbitals. One can see that the Ge and O incorporation introduces impurity states within the bandgap and strong overlap occurs between the states from the Ge 4 $p_z$ and O 2 $p_z$ orbitals, which indicates a strong coupling. Both Ge 4 $p_z$ and O 2 $p_z$ orbitals possess out-of-plane lobe configuration along the z-axis, which could well match to each other spatially, symmetrically (t$_2$ symmetry) as well as energetically. This $pp\pi$ type orbital hybridization, as illustrated in Fig. 1g, leads to strong side-by-side coupling as well as energy level splitting. The splitting forms a bonding $\pi$ orbital (Level-2) in the lower energy side and another anti-bonding $\pi$* orbital (Level-1') in the higher energy side. According to the principle of total energy conservation as well as the level repulsion between different orbitals[19,30], if the $\pi$ level is pulled downward the energy of the $\pi$* level could be pushed up. In this way, by sacrificing the new deeper $\pi$ level, the $\pi$* level is able to get closer to CBM and behaves as an extremely shallow donor level. Figure 1h shows the charge distribution of Level-1' in the Ge-O$_3$ system, which mainly appears on the Ge site in a shape as an anti-bonding $\pi$* orbital. Meanwhile, one can also see that these charges further extend onto the nearby B-O bond in $\pi$ orbitals and even farther B sites in

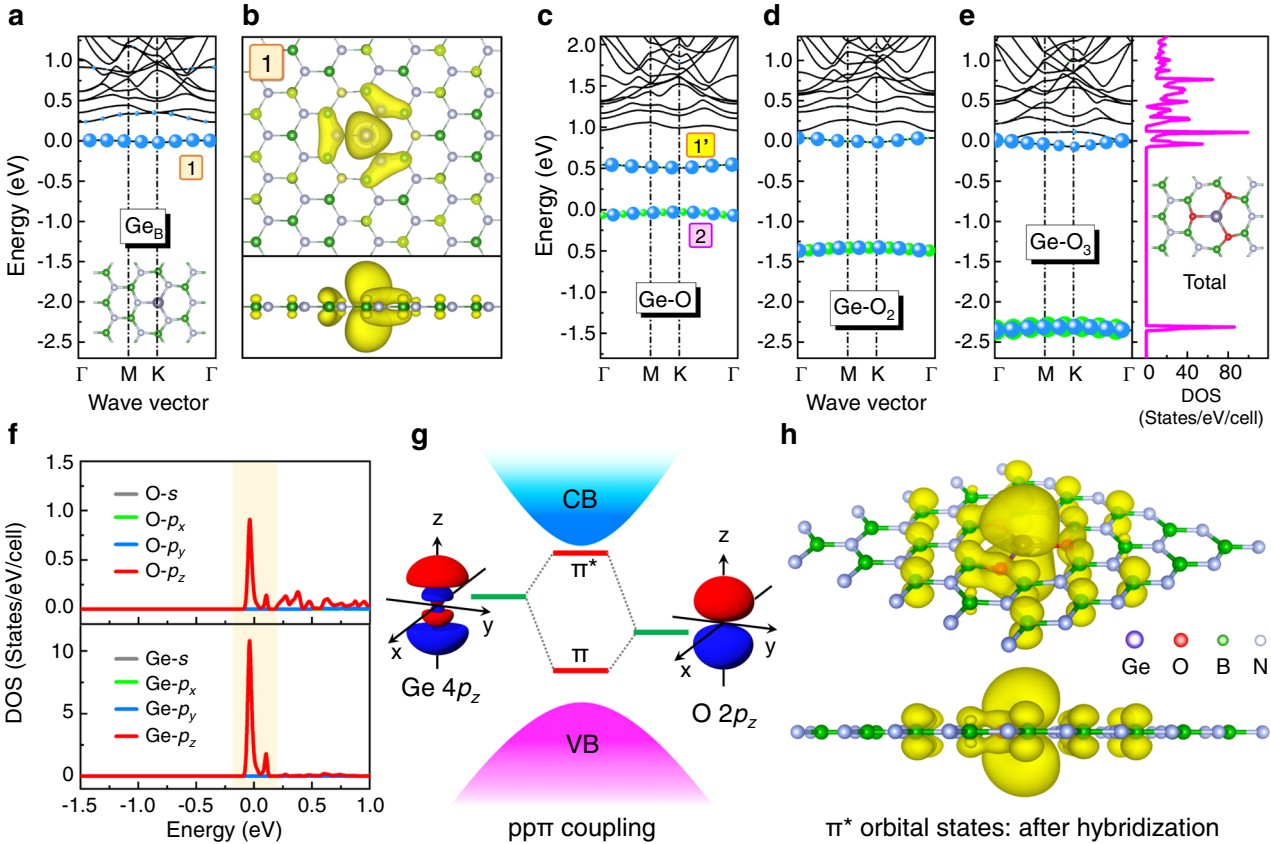

**Fig. 1 Sacrificial impurity coupling induced orbital level engineering. a** Energy band structure of h-BN:Ge_B system. The blue dots represent the Ge-contributed electronic states. The inset shows the atomic structure of Ge doping in the h-BN monolayer. **b** Top view (top) and side view (bottom) charge contributions of the Ge impurity level 1 of h-BN:Ge_B system. **c** Energy band structure of h-BN:Ge-O system. **d** Energy band structure of h-BN:Ge-O_2 system. **e** Energy band structure and total DOSs of h-BN:Ge-O_3 system. The blue and green dots represent the Ge- and O-contributed electronic states, respectively. The inset in total DOSs shows the atomic structure of h-BN:Ge-O_3. **f** Partial DOSs from the Ge and O atoms of h-BN:Ge-O_3 system. **g** Schematic of the orbital split induced level engineering through the $pp\pi$ coupling between the Ge 4 $p_z$ and O 2 $p_z$ orbitals. **h** Top view (top) and side view (bottom) charge contributions of the impurity level 1' of h-BN:Ge-O_3 system.

$p_z$ orbitals. This delocalization feature indicates the preference of ionization of the donor Level-1' into the host lattices for effective in-plane conduction. This is quite different to the localization behavior of the single Ge doping case (Fig. 1b). As we know, the electronegativity of Ge atom (2.01) is smaller than N (3.04) and close to B (2.04). Thus, after the Ge substitution for B, the charge exchange between the Ge dopant and the h-BN host lattice is very limited (Fig. S4a). In contrast, the electronegativity of O atom (3.44) is larger than N. Hence, O could help attract and delocalize those additional electrons away from Ge and improve the in-plane *n*-type conduction in h-BN monolayer (Fig. S4b–d). Of course, some other electrons from O itself are tightly trapped in the deeper π orbital, which has been sacrificed.

**Simultaneous Ge-O incorporation in h-BN monolayer.** Growth of h-BN monolayer modified by sacrificial impurity coupling with Ge-O doping was conducted experimentally with a LPCVD system. The system consists three independent heating zones, as shown in Fig. 2a and Fig. S5. Borazane in the T1 zone was used as the precursor for the h-BN growth and GeO_2 powder in the T2 zone was chosen as the doping source. Meanwhile, the electro-chemically polished Cu foil (Fig. S6a–d)[31] as substrate was placed in the T3 zone. GeO_2 has the melting point at 1086 °C and can provide Ge and O impurities simultaneously by controlling the T2 temperature.

Figure 2a–b shows the overall chemical reaction equations during the processes of h-BN growth and in situ Ge-O doping. The entire heating programs for three zones are shown in Fig. S7. The borazane decomposes into solid ammonia-borane and gaseous borazine by heating up to 96 °C in T1 zone[32]. Then, they are carried by the Ar/H_2 gas flow to the reaction T3 zone through T2 zone. Meanwhile, the gaseous GeO_2 is evaporated in T2 zone, which joins the borazane precursor as a mixture in gas phase. Finally, a two-stage chemical reaction takes place on the catalytic surface of Cu foil. The cross-linking reaction of H-B and N-H groups followed by dehydrogenation leads to the unaligned chain branches[33]. At the same time, the Ge-O_2 trimer could easily bond to these groups and incorporate into the h-BN lattice upon the formation of monolayer.

The sublimation and transportation of GeO_2 are crucial and the heating temperature of the GeO_2 precursor has a great impact on the doping process. We conducted a systematic study on T2 temperature from 600, 700, 800 to 900 °C during the growth. By comparison, the intrinsic undoped h-BN was also grown and characterized (Figs. S8 and S9). As shown in Fig. S10a–d, the increase of the heating temperature of GeO_2 impurity precursor will enhance the supplying dose of GeO_2 and evidently change the shape of h-BN domains. At a low temperature of 600 °C, the triangular-shaped h-BN domains show curved edges and blunt angles, which are different to the normal triangular domain of undoped h-BN (Fig. S8a). This

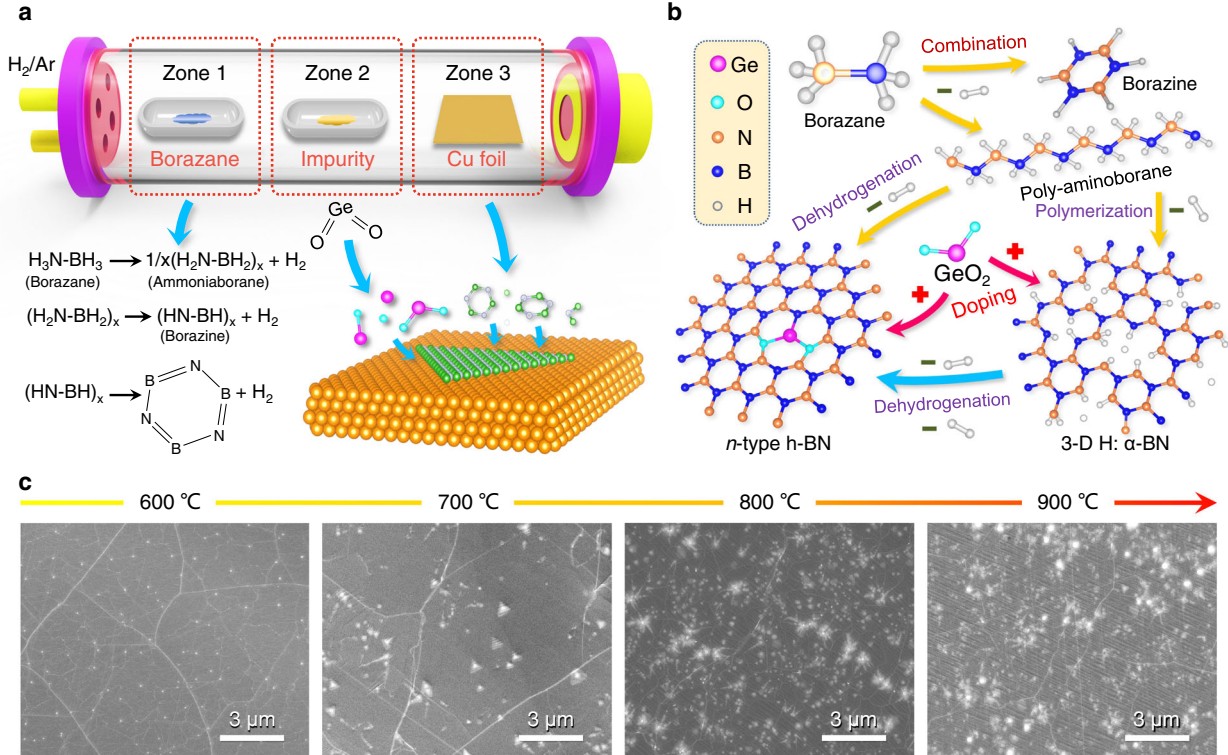

**Fig. 2 Growth of h-BN monolayer using in situ Ge-O doping. a** Schematic of the LPCVD setup for the in situ Ge-O doping in h-BN monolayer. **b** Possible reaction pathways showing the in situ doping of Ge-O and the formation of h-BN:Ge-O monolayer. **c** SEM images of the as-grown h-BN:Ge-O monolayer on Cu foil at various temperatures of zone 2 after 10 min growth times.

could be attributed to the slight impurity doping. When the impurity temperature increases from 700 to 900 °C, the shape of h-BN domains has transferred from triangle to diamond and finally becomes an irregular polygon. It has been reported that the edges of the triangular h-BN domain are nitrogen-terminated, which has lower energy than that of the boron-terminated ones[34,35]. In this condition upon the h-BN:Ge-O growth, the incorporation of Ge, O, or Ge-O into the edge lattice of h-BN domains may break the energy balance between three edges of the triangular h-BN domain. Thus, it will promote the formation of boron-terminated edges and the asymmetric diamond/polygon shapes. Except the shape, the nucleation density of h-BN is also affected by Ge-O doping. As shown in Fig. S11, when the temperature increases to 800 and 900 °C, the nucleation density of h-BN has greatly increased, which is a disadvantage for the growth of high-quality h-BN.

Scanning electron microscope (SEM) images of the fully coalesced h-BN:Ge-O monolayer under various T2 temperatures are shown in Fig. 2c. The full coalescence of domains and the formation of a complete h-BN monolayer can be confirmed by an antioxidant test (Fig. S12). One can observe that at 600–700 °C, the wrinkles and the cleanness on the h-BN film indicate the fully coalesced monolayer and well-preserved crystal quality upon Ge-O incorporation. However, when the T2 temperature further increases up to 800 and 900 °C, the quality of h-BN film observably deteriorates with the presence of large particles. This indicates that the evaporation rate of GeO₂ precursor becomes larger than the incorporation rate of impurity into the h-BN lattice. The excess supply of GeO₂ leads to the deposition of the by-products in form of particles on the surface. Thus, it can be concluded that 700 °C is the optimal T2 temperature for efficient Ge-O doping without affecting the structural quality of h-BN.

**Confirmation of Ge-O incorporation in h-BN monolayer**. The as-grown h-BN:Ge-O film was transferred onto a SiO₂ substrate using the PMMA-assisted method (Figs. S13 and S14). Figure 3a shows the atomic force microscope (AFM) image of the edge area of the h-BN:Ge-O layer. The thickness is measured to be about 0.59 nm, indicating the monolayer of h-BN. The film was transferred onto a transmission electron microscopy (TEM) grid by PMMA-free technique (Fig. 3b) and the high-resolution TEM (HRTEM) image was obtained (Fig. 3c). The clear hexagonal atomic structure can be observed. Meanwhile, the selected area electron diffraction (SAED) shows a bright hexagonal diffraction pattern of (10-10) index of h-BN, indicating the high crystal quality of h-BN:Ge-O monolayer (Fig. 3d). Detailed examination of the elemental contrast in Fig. 3c reveals the existence of darker contrast sites in this hexagonal pattern (marked by red circles). In principle, the larger atomic radius of Ge and O will lead to the dark contract, which confirms the substitution of Ge-O for B and N atoms.

To further confirm the incorporation of Ge-O into the h-BN lattice, transmission and absorption spectra were detected for the undoped and Ge-O doped h-BN films on transparent sapphire substrates (Fig. 3e and Fig. S15). One can see that a distinct band-edge absorption peak locates at 5.93 eV for both samples. While, an additional shoulder appears at the lower energy side of the band-edge peak for the h-BN:Ge-O film, which should be attributed to the Ge-O impurity level related absorption (see Fig. S16 and related discussion). Figure 3f shows the Raman spectrum of the h-BN:Ge-O film, which can be compared with those of the undoped h-BN and Si substrate (Fig. S17). In contrast to the typical Raman spectrum of E₂g vibrational mode (1369.5 cm⁻¹) of the undoped h-BN monolayer (Fig. S18), the spectrum of h-BN:Ge-O exhibits an asymmetric line shape with two decomposed peaks located at

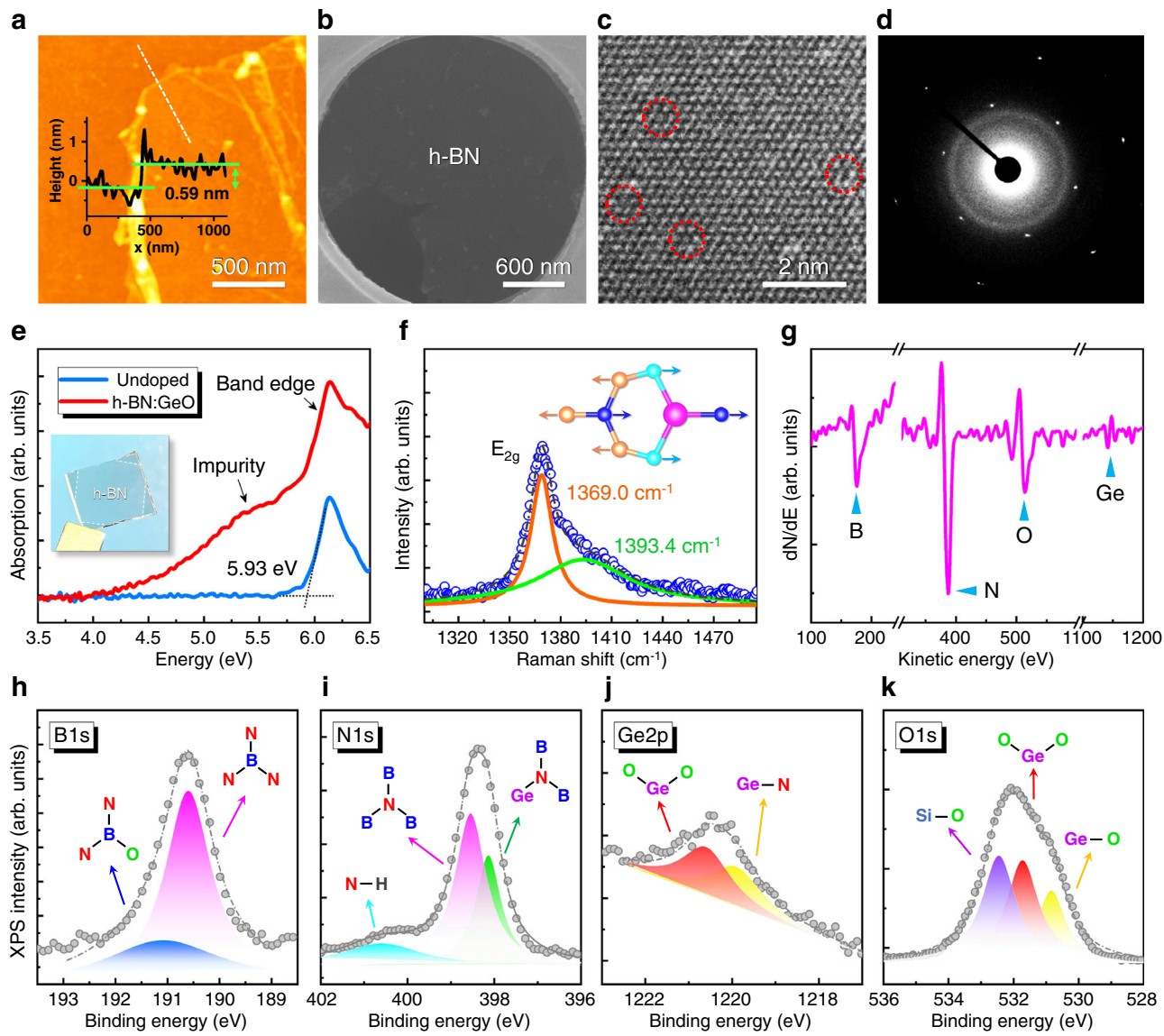

**Fig. 3 Structural characterization of the h-BN:Ge-O monolayer. a** AFM image of the h-BN:Ge-O film on SiO$_2$, showing monolayer thickness. **b** SEM image of the suspended h-BN:Ge-O monolayer supported on a TEM grid. **c** HRTEM image and (**d**) SAED pattern of the h-BN:Ge-O monolayer. **e** Absorption spectra of the undoped h-BN and h-BN:Ge-O. **f** Raman spectrum of the h-BN:Ge-O film. The inset shows the corresponding E$_{2g}$ vibrational mode. **g** AES spectrum of the h-BN:Ge-O film. **h**–**k** XPS spectra of the h-BN:Ge-O film, showing the B $1s$, N $1s$, Ge $2p$, and O $1s$ core levels, respectively.

1369.0 and 1393.4 cm$^{-1}$. The 1369.0 cm$^{-1}$ main peaks can be assigned to the in-plane E$_{2g}$ vibration mode of hexagonal B–N bonds[36]. The shoulder peak at 1393.4 cm$^{-1}$ can be assigned to the shifted local vibration mode affected by the Ge-N and O-B bonds, suggesting the incorporation of Ge and O into the h-BN lattice. The chemical compositions of the h-BN:Ge-O film was further investigated by the Auger electron spectroscopy (AES), as shown in Fig. 3g. Besides the intense peaks from B$_{KLL}$ and N$_{KLL}$ Auger lines, the signals from O (514.1 eV) and Ge (1144.7 eV) also can be detected[37], further confirming the existence of Ge-O in h-BN layer.

To verify the chemical bonding between Ge and O atoms, X-ray photoelectron spectroscopy (XPS) measurements on B, N, Ge and O elements were carried out. Figure 3h–i shows the spectra of B $1s$, N $1s$, and their decomposed components. The B $1s$ peak consists of two decomposed peaks at 190.6 and 191.2 eV, which can be assigned to the electrons from B-N bond and B-O bond, respectively[38]. This indicates the incorporation of O as a substitutional dopant for N. The N $1s$ peak exhibits fitted

components from N-H (400.6 eV), N-B (398.5 eV), and N-Ge (398.1 eV), confirming the Ge$_B$ doping in h-BN[39]. Regarding the Ge-O interaction, Fig. 3j–k show the Ge $2p$, O $1s$ peaks and their decomposed spectra. The components at 1219.9 and 1220.6 eV in the Ge $2p$ peak are contributed from Ge-N and Ge-O$_2$, respectively[40,41]. This fact confirms both the incorporation of Ge in h-BN as well as the trimer bonding with O. In addition, the Ge $3d$ core level was also investigated, which consistently shows the corresponding Ge-O$_2$ (32.6 eV) and Ge-N (32.0 eV) peaks[40,42] (Fig. S19). From the spectrum of O $1s$ in Fig. 3k, one can see the Ge-O$_2$ (531.7 eV) and Ge-O (530.8 eV) components clearly, confirming the incorporated dimer and/or trimer Ge-O in the h-BN lattice. Since the doping level is a crucial parameter for the $n$-type h-BN, XPS measurement was employed to estimate the Ge-O doping concentration. The substitutional doping level of Ge$_B$ is about 2.8%, which indicates a heavy doping and is consistent with the absorption spectrum (Fig. 3e). The actual doping concentration could be overestimated by XPS due to the ultrathin thickness of h-BN monolayer. The absorption spectrum

of the h-BN:Ge-O showed an impurity peak and a dominant band edge peak at ~5.93 eV, consistent with that of the intrinsic h-BN (Fig. 3e). This indicates that the Ge-O is incorporated into h-BN as dopants instead of forming a new alloy. By comparison, the absence of Ge in the undoped h-BN was also confirmed (Fig. S20). Furthermore, based on the recorded XPS data, the chemical stoichiometric ratio between B and N has been determined to be 1/0.88. This indicates that the substitution of N by O is more than the substitutional B by Ge, referring to the mixed doping configurations of $Ge-O_2$ or $Ge-O_3$ rather than merely the Ge-O dimer form. Moreover, based on the first-principles calculations, the decreasing formation energy follows the trend of $Ge_B > Ge_B-O_N > Ge_B-2O_N > Ge_B-3O_N$. Therefore, the $Ge-O_2$ trimer and $Ge-O_3$ tetramer are also energetically favorable in the thermodynamics aspect (see Fig. S21 and related discussion). Summarized from the above results, the incorporation of Ge and sacrificial O dopants in h-BN monolayer makes their existence in the coupled forms of Ge-O, $Ge-O_2$ or $Ge-O_3$, which is consistent with the previous theoretical simulation results.

**Efficient *n*-type conductivity of h-BN:Ge-O monolayer**. The h-BN:Ge-O monolayer film was transferred onto a *n*-Si substrate (Fig. S22) and the conduction AFM (CAFM) were conducted to measure the through-plane conductivity (the inset of Fig. 4a). Undoped h-BN was taken as the control sample for comparison. The measurement was carried out under a forward voltage of 5 V and the line scan and mapping of the through-plane current are shown in Fig. 4a–c. Figure 4a gives the line scan on the undoped and Ge-O doped h-BN monolayers. One can see that the undoped h-BN monolayer exhibits a highly insulating property (~pA) whereas the through-plane current in several nA can only be detected at some local points, which should originate from point defects such as $V_B$ or $C_B$. The distribution of a few conducting point defects can be confirmed by the mapping image in Fig. 4b. Such strongly localized current injection points could also provide high-quality single-photon emitters[43,44]. In contrast, the h-BN:Ge-O monolayer has a significant through-plane current in the order of 100 nA, as illustrated in Fig. 4a (the lower panel). Meanwhile, this conduction has covered the entire area of h-BN film, as shown in the current mapping of Fig. 4c. These evidences strongly demonstrate the efficient doping of Ge-O impurities and the effective thermal ionization for through-plane conduction, as proposed by the DFT simulations. To test the in-plane conduction of h-BN:Ge-O monolayer, the film was transferred onto insulating sapphire substrates with pre-fabricated Au electrode array (Fig. 4d and the inset of Fig. 4e). From the I-V curve shown in Fig. 4e, one can observe that the in-plane current of h-BN:Ge-O reaches ~20 nA under 15 V, showing a good 2D conductivity. For comparison and confirmation of the crucial role of the sacrificial Ge-O trimer, h-BN samples with single dopants (Ge or O) were prepared and studied (Fig. S23 and S24). The results showed that the undoped h-BN and the h-BN:Ge are completely insulating with in-plane current in order of ~pA and the h-BN:O only has very weak in-plane current (~0.5 nA) (Fig. S24a–e). These facts verify that the Ge-O coupling is necessary for the formation of shallow donor levels through sacrificial impurity (O) bonding. The shallow donor could then be effectively activated to provide additional electrons into the host lattice and enhance the in-plane conduction. In addition, because the shallowed Ge-O donor level has lower activation energy, regular band conductivity should be dominant for h-BN:Ge-O at room temperature. However, due to the heavy doping of Ge-O (>2.8%), the conduction mechanism could be partially contributed by hopping conductivity[45,46].

It is of importance to confirm the *n*-type conductivity of h-BN:Ge-O monolayer experimentally. First, ultraviolet photoelectron spectroscopy (UPS) measurements were conducted to measure the work function. As shown in Fig. 4f, by using the intersection-point method[47], the work function of h-BN:Ge-O film could be determined to be about 4.1 eV. Previous literatures have reported that the work function of intrinsic h-BN is in the range between 6.4 and 7.8 eV due to the ultra-wide bandgap[48–50]. The lower work function of h-BN:Ge-O corresponds to the upward shift of the Fermi level closer to CBM, indicating the transition from intrinsic h-BN into *n*-type conduction. Ohmic contact of electrodes is not well obtained due to the work function difference between Au electrode (5.1 eV) and *n*-type h-BN (4.10 eV), which makes it difficult for Hall measurements. Alternatively, a field-effect transistor (FET) device using the h-BN:Ge-O monolayer as the channel layer was fabricated, as illustrated in Fig. 4g. The output characteristic of the FET device with gate voltage ranging from −10 to 10 V and the transfer characteristic with a constant source-drain voltage of 20 V were measured, as illustrated in Fig. 4h–i, respectively. One can see that the source-drain current increases with the gate voltage increasing, which suggests that the electron is the majority carrier in the channel layer. This clearly confirms the *n*-type channel behavior[51]. Electrical properties of the *n*-type h-BN monolayer were further obtained by FET device method[52] (see details in Supplementary Information). The electron mobility and concentration of *n*-type h-BN are determined to be $0.014\,cm^2\,V^{-1}\,s^{-1}$ and $1.94 \times 10^{16}\,cm^{-3}$, respectively. Thus, it could be convinced that by applying the sacrificial impurity coupling method via in situ Ge-O doping, we have successfully obtained efficient *n*-type conductivity in the h-BN monolayer.

**Vertically stacked *p-n* junction with *n*-hBN/*p*-GaN**. In previous work, the Schottky diodes based on undoped h-BN/GaN were reported[53]. The realization of *n*-type h-BN provides the possibility to fabricate advanced optoelectronic devices beyond the conventional group-III nitride material system. As shown in Fig. 5a–b, we construct a vertically stacked heterojunction with *n*-hBN and *p*-GaN by transferring the h-BN layer in various thicknesses (monolayer to multilayers) onto the surface of *p*-GaN. Based on the material parameters of *p*-GaN and *n*-hBN, including bandgap, electronic affinity, and work function[28,54–57], the band configuration of the *p-n* junction is drawn in Fig. 5c. The band alignment between *p*-GaN and *n*-type h-BN shows a Type-I heterostructure and the conduction and valence band offsets are 1.9 and 1.2 eV, respectively. Figure 5d–e demonstrates the I-V curves of the *p-n* junction with *n*-type h-BN in a thickness of 1 and 6 monolayers, respectively. One can see the typical rectification behavior of a diode, which further confirms the *n*-type conductivity of h-BN and the *p-n* junction structure. For the one monolayer case, the forward voltage is about 1.44 V and the reverse leakage current could reach 4.01 μA under −8 V. The reverse leakage current, in principle, is related with the diffusion of minority carriers which are holes in this *n*-type h-BN. Since the h-BN has an intrinsic background carrier of *p*-type holes induced by the formation of B vacancies during high-temperature growth, the reverse leakage current is still considerable. And the rectification ratio between ±8 V is only 2.6 (Fig. 5d). Meanwhile, for this type of vertically stacked 2D diode, this leakage current is also contributed by the tunneling current through the ultra-thin h-BN layer. From this point of view, the leakage could be minimized by increasing the thickness of h-BN with multilayer stacking. Thus, when the 6-monolayer *n*-type h-BN is used (Fig. 5e), the leakage current has been reduced down to only 0.08 μA and the rectification ratio is enhanced up to 167.7, showing

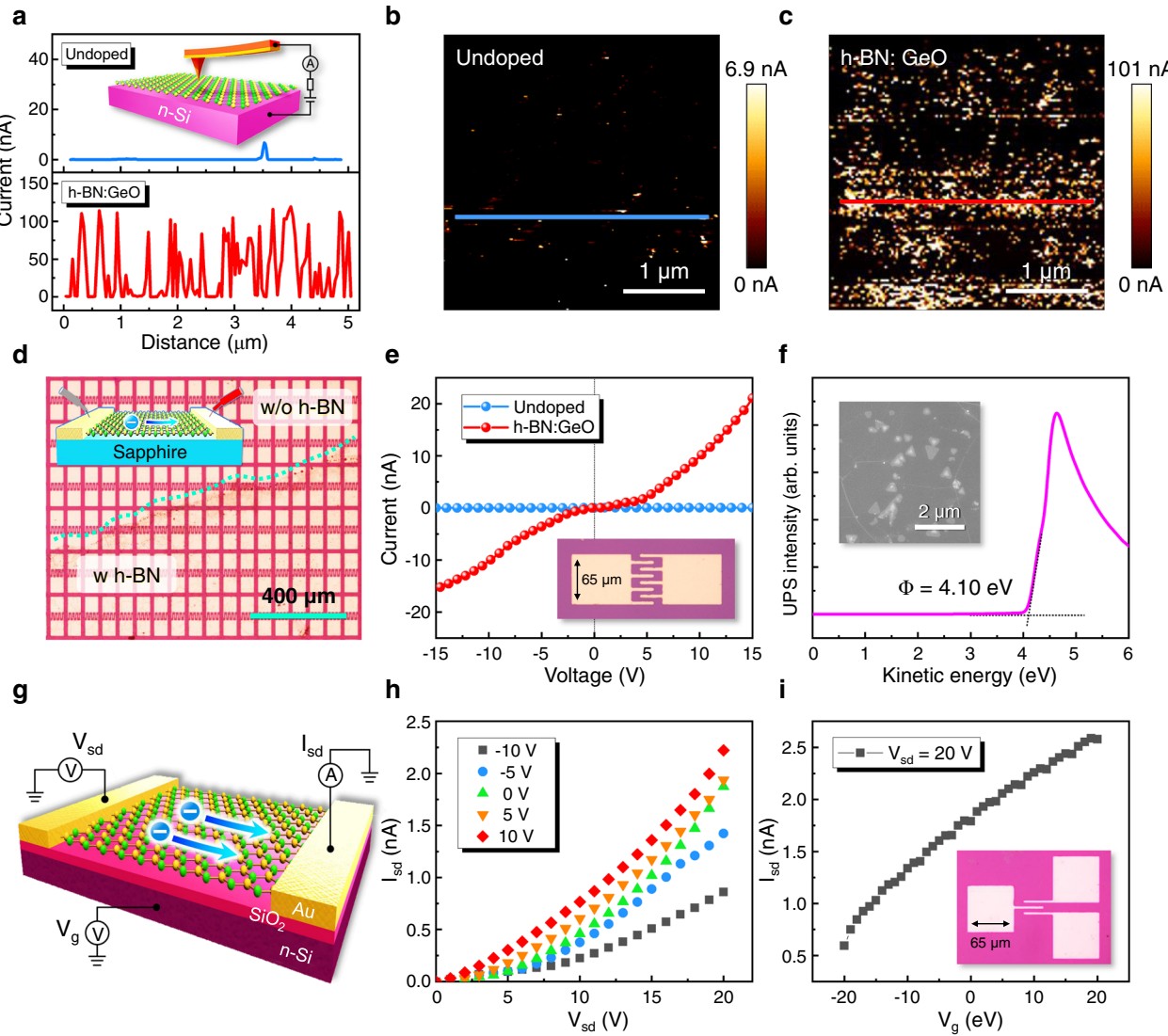

**Fig. 4 Electrical properties of the h-BN:Ge-O monolayer. a** Current line scan of CAFM test for undoped h-BN and h-BN:Ge-O, respectively. The inset shows the setup of the CAFM. **b–c** The current mapping of CAFM test for the undoped h-BN and h-BN:Ge-O, respectively. **d** Optical image of h-BN monolayer transferred on sapphire substrate with Au electrode array. **e** I–V curve of the h-BN:Ge-O monolayer compared with the undoped h-BN. **f** UPS spectrum of the h-BN:Ge-O monolayer in the kinetic energy scale. Work function is determined by the intersection-point method. **g** Schematic of the FET devise using h-BN:Ge-O monolayer as $n$-type channel. **h–i** Output characteristics and transfer curve of the h-BN:Ge-O monolayer based FET device.

an excellent rectification characteristic. Moreover, the forward voltage increases from 1.44 eV for 1-monolayer $n$-hBN device to 3.11 eV for the 6-monolayer one. The underlying mechanism is rooted in the depletion region and the built-in electric field. For the 1-monolayer $n$-hBN device, such an ultrathin $n$-type h-BN layer should be completely depleted and the built-in field will decrease with the decreasing h-BN thickness (see Fig. S26 and related discussion). Thus, the forward turn-on voltage for the monolayer h-BN case is smaller than that of the 6-layer $n$-h-BN case.

Capacitance-voltage (C-V) measurements were further carried out on the device, as shown in Fig. 5f. One can see that the reverse-biased $p$-$n$ junction shows the monotonically decreasing capacitance with the voltage increasing, mainly due to the depletion width increasing. As we know, the $p$-$n$ junction capacitance is contributed by two types, the diffusion capacitance, and the transition capacitance[58]. While, in the reverse-biased diode, the transition capacitance dominates whereas the diffusion capacitance remains constant because the amount of minority carriers (holes) in $n$-type h-BN is very low. Therefore, the

decreasing transition capacitance indicates the widening of the depletion width, which is highly related with the increasing thickness of the $n$-type h-BN layer. In contrast, in the forward-biased diode, one can observe that the capacitance keeps increasing with the increasing voltage in rang I (0~1.44 V), which reflects the space charge accumulation process at the depletion region. This indicates the dominance of the diffusion capacitance with the charge storage. After the peak position (1.44 V), the capacitance begins to decrease mainly due to the narrowing of the depletion region width upon the recombination by the majority of carriers (electrons from $n$-hBN). When the modulated frequency of the $ac$ signal increases from 3 MHz up to 5 MHz, we find a systematic increase in the total capacitance while the diffusion capacitance (>1.44 V) drops steeply under an $ac$ signal of 5 MHz. The phenomenon of increasing capacitance with the increasing frequency is unusual for a $p$-$n$ junction. This may stem from the existence of a considerable inner series inductance[59] within $n$-hBN layers, which is determined to be about 0.0035 H (See details in Supplementary Information,

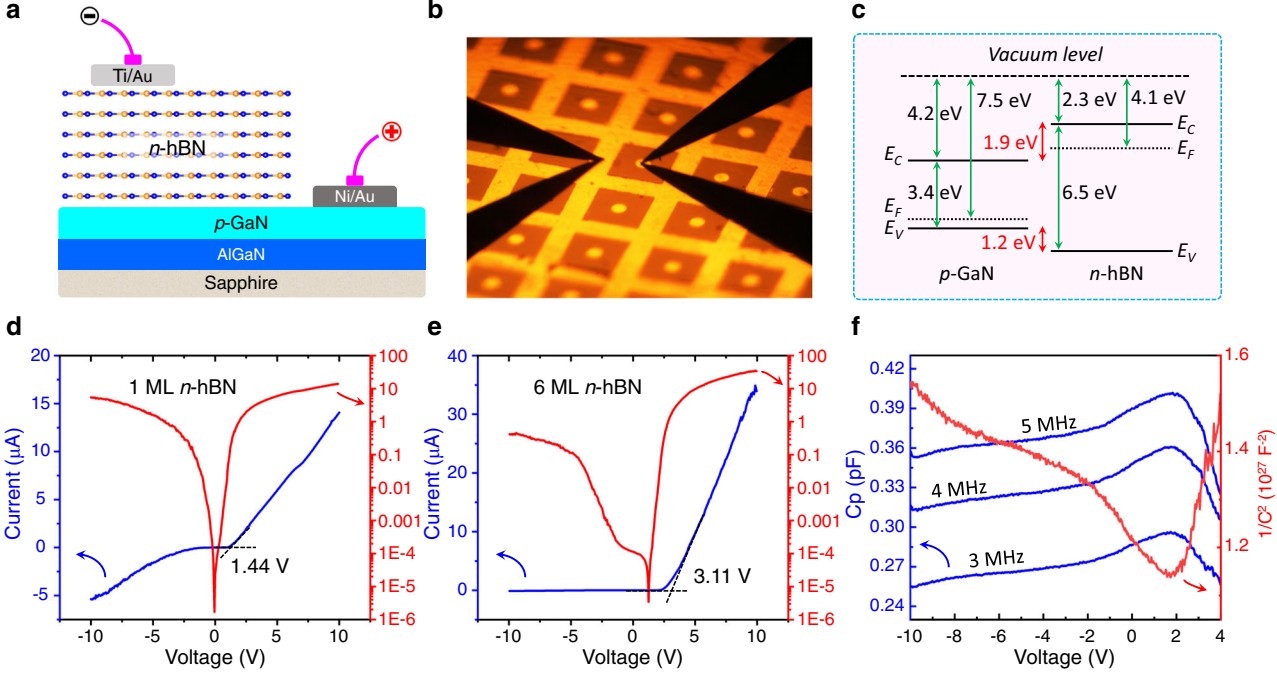

**Fig. 5 Vertically stacking heterojunction based on the *n*-hBN/*p*-GaN. a** Schematic of the *p-n* junction based on *n*-type h-BN and *p*-type GaN. **b** Photograph of the fabricated *n*-hBN/*p*-GaN junction array on board. **c** Band diagram of the *n*-hBN/*p*-GaN junction. **d–e** Linear and logarithmic I-V curves of the *p-n* junction based on one monolayer and six monolayers *n*-type h-BN, respectively. **f** C-V characteristics measured from reverse to forward bias of the 6-monolayer *n*-hBN/*p*-GaN junction under various *ac* frequencies and corresponding $1/C^2$ plot versus voltage under a frequency of 3 MHz.

Fig. S27). In addition, the $1/C^2$-V curve is shown in Fig. 5f, which appears nonlinear at zero voltage. The nonlinear feature of $1/C^2$-V curve can be attributed to the inhomogeneous doping profile in the interfacial region of *n*-hBN/*p*-GaN. It's worth noting that the total capacitance of this *n*-hBN/*p*-GaN junction is as small as ~pF due to the ultrathin 2D h-BN/3D GaN integration. This ultra-small capacitance is quite beneficial for enhancing the response speed of optoelectronic devices, e.g. photodetectors[60,61]. These decent *p-n* diode behaviors promise the potential fabrication of advanced optoelectronic devices by the heterogeneous integration of conventional 3D nitride epilayer and novel 2D h-BN conducting layer.

In conclusion, we proposed a novel strategy of orbital split induced level engineering through sacrificial impurity coupling to achieve effective *n*-type conduction in monolayer h-BN. To overcome the doping asymmetry limitation of ultra-wide bandgap h-BN, an extremely shallow donor level was designed through the coupling of Ge 4 $p_z$ and O 2 $p_z$ orbitals. Due to the same $t_2$ symmetry and wavefunction character, these two orbitals can strongly couple with each other. The introduction of O side-by-side to Ge donor can effectively push up the donor level by the formation of another sacrificial deep level. The in situ Ge-O doping in h-BN monolayer was successfully achieved by employing $GeO_2$ as the impurity precursor with a LPCVD system. In contrast to the high insulation of the intrinsic, individual Ge-doped and O-doped h-BN, the Ge-O trimer doped h-BN monolayer demonstrated an efficient *n*-type conductivity with a considerable through-plane (~100 nA) and in-plane (~20 nA) current. A decent diode behavior in the vertically stacked heterojunction of *n*-type h-BN and *p*-type GaN with a large rectification ratio of 167.7 was achieved. The sacrificial impurity coupling method will expand application scenarios of advanced optoelectronic and electronic devices based on the bipolar conducting h-BN and other 2D/3D materials.

## Methods

**Theoretical calculations**. All the first principle simulations were carried out using Vienna ab-initio simulation package (VASP) in the framework of DFT[62,63]. The generalized gradient approximation (GGA) with the Perdew–Burke–Ernzerhof (PBE) function was implemented for the exchange-correlation interactions among the electrons[64]. A cutoff energy of 550 eV was used to expand the electronic wavefunctions and a $9 \times 9 \times 1$ Monkhorst-Pack grid of *k* points was used for sampling the Brillouin zone. The monolayer h-BN supercell generated by $5a \times 5b \times 1c$ primitive cells was adopted (Fig. S3). A vacuum layer of about 20 Å was applied, which was determined to be sufficiently large to avoid interaction between neighboring supercells. The geometric optimization was performed by relaxing all the degrees of freedom using the conjugate gradient algorithm until all forces became <0.01 eV/Å.

**Epitaxial growth**. The large-scale h-BN monolayer was epitaxial grown using a LPCVD system that consists of three independent heating zones (Fig. S5). A 25 µm thick Cu foil (Alfa Aesar, product no. 13382) was used as substrate and placed in the reaction zone 3. The Cu foil was electrochemically polished under a 4 A current to remove the oxides and improve the surface flatness before loaded into the quartz tube chamber (Fig. S6). Borazane (Toronto Research Chemicals) was used as the precursor of B and N, and placed in zone 1. Germanium (Ge, 99.999%, 9Ding-Chem) and Boron Oxide ($B_2O_3$, 99.0%, Maikun Chemical) was used as the precursor of Ge and O for the conventional individual doping, and Germanium dioxide ($GeO_2$, 99.999%, Macklin Biochemical) was used as the impurity precursor for the in situ Ge-O doping in h-BN, respectively. The impurity precursor of about 5 mg was placed in zone 2 and independently heated up to a temperature around the melting point during the growth of h-BN. A mix of $H_2$ and Ar gases was used as the carrier gas. To remove the oxides and enlarge the grain size of Cu foil, the annealing process was carried out at 1000 °C under an Ar/$H_2$ gas flow for 30 min. After that, the T1 precursor zone and T3 growth zone were ramped up to 96 and 1050 °C, respectively. Meanwhile, the evaporation of borazane and $GeO_2$ were carried by the Ar/$H_2$ gas flow into zone 3. Except the zone 2, all growth conditions for the intrinsic, individual Ge-doped, O-doped, and Ge-O doped h-BN were fixed in the same. After the growth, the Cu foil and borazane were pulled out from zone 1 and zone 3 immediately by a quartz manipulator with a magnetic slider for rapid cooling. Finally, the chamber was cooled down to room temperature slowly and the sample was kept protected by the gas flow. Details of the growth process can be found in Fig. S7. The *p*-GaN was grown above an AlGaN template and the thickness of *p*-GaN is about 20 nm. The Mg-doping concentration is at the level of $10^{19}$ cm$^{-3}$ and the resistivity is about 6.27 Ω cm in the *p*-GaN (Fig. S25). The size of the *p-n* junction device is $400 \times 400$ µm.

**Characterization.** The morphology of h-BN was characterized by using a scanning electron microscope (SEM, Hitachi S-4800). The monolayer thickness and vertical conduction of h-BN films were measured by a CAFM system (Bruker Nano GmbH, Bruker NW4). For the antioxidant test, the bare Cu foil and Cu foil covered with fully coalesced h-BN monolayer were placed on a heating plate, then the samples were heated in air at 200 ºC for 10 min. Light transmission and absorption spectra were collected by a UV-Vis-NIR spectrophotometer (Agilent Technologies Cary 50000). Raman spectra were collected by a Raman microscope (WITec alpha 300RA) with a 488 nm laser. The X-ray photoelectron spectroscopy (XPS, PHI Quantera) and the Auger electron spectroscopy (AES, PHI660 system) were conducted to confirm the composition and impurity doping in the h-BN monolayer. UPS (Thermo Fischer, ESCALAB Xi$^+$) was employed to determine the electronic work function of h-BN films. The transmission electron microscopy (TEM) investigation was carried out using a FEI Talos F200s field-emission electron microscope. The electrical measurement of I–V curve was carried out on a probe station with a Keithley 2450 system. The capacitance-voltage measurement was conducted in a semiconductor parameter analyzer (Tektronix, 4200A-SCS).

## Data availability

The data that support the findings of this study are available from the corresponding authors upon reasonable request.

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

## Acknowledgements

This work was partly supported by the National Key Research and Development Program (2016YFB0400800), the National Natural Science Foundation (62074133, 61974124, 62135013, and 11804115), and the Science and Technology Programs of Fujian Province (2021H0001) of China.

## Author contributions

S.Q.L. and P.S. contributed equally to this work and should be considered as co-first authors. S.Q.L., P.S., and D.J.C. conceived the idea, designed the experiments and wrote the manuscript. S.Q.L. and P.S. performed first-principles simulations. S.Q.L. and H.Y.Z. performed the epitaxial growth. S.Q.L., G.Z.L., and B.G. carried out the device fabrication. S.Q.L., G.Z.L., Y.H.C., H.C., and F.Y.X. performed characterization and measurement of all samples. F.Y.X., T.C.Z., F.C.X., X.H.C., and J.Y.K. participated in data analysis. All authors discussed the results and commented on the manuscript.

## Competing interests

The authors declare no competing interests.
