## [Peer Review File · Nature Communications]

REVIEWER COMMENTS

Reviewer #1 (Remarks to the Author):

In the paper "Hexagonal Boron Nitride Can Obtain n-Type Conductivity" by Shiqiang Lua et al., the authors investigated n-type doping in h-BN both theoretically and experimentally. This topic is important for the development of h-BN-based optoelectronic devices and is well motivated by the authors in the introduction. Obtained results convince that a Ge-O₂ trimer brings the shallow donor level with relatively low ionization energy. This concept is new and may be interesting to publish in Nature Communications, but I have a few points that need to be addressed prior to publication:

1. On page 4 lines 15-17 the authors claim "According to the principle of total energy conservation, the energy of the π^* level could be pushed up if at the same time the π level is pulled downward." This statement is not obvious to me and should be supported by appropriate references or other arguments.

2. On page 12, lines 1-3 the authors claim "The lower work function of h-BN:Ge-O indicates the upward shift of the Fermi level of h-BN:Ge-O closer to the CBM, demonstrating the n-type nature." The conclusion about the n-type nature is too strong considering the results of the UPS measurements and therefore this statement should be revised.

3. The authors do not comment on the Ge-O doping level, while it is well known that this is a key parameter in the optimal doping. The other question related to this issue is the character of conductivity. Is it hopping conductivity or regular electron conductivity in the conduction band?

4. What about Ge and O concentration in the reference samples?

5. The strong absorption at 250 nm, see Fig.3 (e) and S15, seems to be very important and is too shortly commented in the paper. What is the origin of this absorption? Which impurity level is responsible for this absorption? What about acceptor-like states in these samples and their compensation by donor-like states?

6. n-hBN/p-GaN structure was studied, but details on p-type layer (level of Mg doping etc.) are not provided. Moreover, this structure and measurements on this structure are not commented on in

the context of previous research for h-BN/GaN structures, see ACS Applied Materials & Interfaces 2022, 14, 4, 6131-613.

7. Is there a “driving force” other than statistics that causes Ge to bind to O? This issue should be better commented on in this article.

8. Why are Hall's measurements not made on these samples? Are there problems with ohmic contacts to h-BN? This issue should be commented on in this article.

Reviewer #2 (Remarks to the Author):

This work reports on a study aiming at demonstrating the n-type doping of h-BN epitaxial films. The latter are of importance and are the subject of strong interest due to their potential for various applications, such as 2D heterostructures and deep UV LEDs and lasers. Efficient doping of wide bandgap materials is challenging. This is especially the case of h-BN for which it is not completely clear that both p and n doping have been achieved despite numerous (and contradictory) theoretical and experimental works. The paper provides an impressive and almost complete bundle of theoretical and experimental results contributing to the demonstration of n-doped h-BN.

The approach followed by the authors is very convincing and significant. Using first-principles calculations, the authors first investigated the strong orbital coupling between donor impurity atom and various coordinating atoms leading to the identification of Ge-O dimer, trimer and tetramer as efficient donors. They then grew the corresponding epitaxial films using LP-CVD technique. HRTEM, SAED, Raman, AES, XPS were then used to confirm the chemical doping. Finally, electrical measurements on h-BN films and p-GaN/n-BN were carried out to assess the electrical doping. Such a deep and rather complete approach is completely original for the study of n-doping in h-BN although similar approach has been used for p-doping.

There is no doubt that the chemical doping has been achieved and that the conductivity has been altered by the Ge-Ox incorporation in the h-BN lattice. The evidences are clear. Nevertheless, Hall measurements would have been significant and very useful to estimate both the doping concentration and carrier mobility. Also, since p-BN is achievable why have the authors chosen the fabrication of a n-BN/p-GaN heterostructure instead of a p-BN/n-BN homojunction?

In addition, some points require more clarification:

- C-V measurements: Fig.5-f shows the C-V for different AC frequencies. It is reported an increase of the capacitance with the frequency. This is unexpected. The capacitance should not depend on frequency except in presence of defect which would tend to decrease it. What is the origin of such a phenomenon? At large reverse voltage, no saturation of the capacitance is observed. Does it mean that the n-doping is weak? The $1/C^2=f(V)$ plot would have been useful to confirm the voltage threshold of the diode.

- I-V measurements: The size of the devices is not indicated. The doping of p-GaN is not given. It is probably much larger than that of n-h-BN. Thus, the n-h-BN layer should be completely depleted (for sure for the single layer and probably for the 6-multi-layers also). How does it impact the diode behavior?

- h-BN monolayer: Its thickness is said to be 0.59 nm. This value is twice the one found in the literature. Why?

- p-doping of h-BN: It is said that it can be easily achieved. It seems that it is not the case. Some controversy still exists, and the p-doping is not very efficient.

- Others: Page 8, line 1: Could you define what an antioxidant test is? Fig.4-c: It is said that the conductivity is uniform, but it does not look like that. Last, the title of the paper is a little weird and the English language requires improvements.

Overall, this paper deserves publication after mandatory revisions described in the previous paragraph and strong improvement of the English language.

Reply to the report of **Reviewer #1** on NCOMMS-22-05830-T/Lu et al.

- **Reviewer #1 wrote:** In the paper “Hexagonal Boron Nitride Can Obtain n-Type Conductivity” by Shiqiang Lu et al., the authors investigated n-type doping in h-BN both theoretically and experimentally. This topic is important for the development of h-BN-based optoelectronic devices and is well motivated by the authors in the introduction. Obtained results convince that a Ge-O₂ trimer brings the shallow donor level with relatively low ionization energy. This concept is new and may be interesting to publish in Nature Communications, but I have a few points that need to be addressed prior to publication.

Authors reply:

We sincerely thank the Reviewer for his/her careful reading of our paper, providing helpful comments, and especially the kind recommendation for the publication of our manuscript. We have considered all the suggestions and revised the manuscript accordingly.

- **Reviewer #1 wrote:** 1. On page 4 lines 15-17 the authors claim “According to the principle of total energy conservation, the energy of the π^* level could be pushed up if at the same time the π level is pulled downward.” This statement is not obvious to me and should be supported by appropriate references or other arguments.

Authors reply:

We thank the Referee for the helpful suggestion. We would like to further explain this issue in more details. Upon the orbital coupling, taking $pp\pi$ coupling as an example (**Figure R1**), the coupling of two matched p_z orbitals will lead to the formation of new split orbitals (π and π^*). In principle, the energy level splitting is induced by the strong coupling, hybridization, and the level repulsion between different orbitals. After the release of the binding energy, the total energy of orbitals under coupling should be conserved. Therefore, if the energy of the π orbital becomes even lower, the energy of π^* will shift up to a higher energy [*Physical Review Letters*, 98, 135506 (2007)] [*Applied Physics Express*, 6, 042104 (2013)].

According to the Reviewer’s suggestion, the statement has been revised as “According to the principle of total energy conservation as well as the level repulsion between different orbitals, if the π level is pulled downward the energy of the π^* level could be pushed up.” References were added accordingly.

Figure R1. Schematic of energy levels of $pp\pi$ orbital coupling and splitting.

Actions taken:

We have revised:

- (1) **Line 14, Page 5**, “According to the principle of total energy conservation, the energy of the π^* level could be pushed up if at the same time the π level is pulled downward.” → “According to the principle of total energy conservation as well as the level repulsion between different orbitals,^{19,30} if the π level is pulled downward the energy of the π^* level could be pushed up.”

We have added:

- (2) **References**

[*Physical Review Letters*, 98, 135506 (2007)] as reference [19].

[*Applied Physics Express*, 6, 042104 (2013)] as new reference [30].

- **Reviewer #1 wrote:** 2. On page 12, lines 1-3 the authors claim “The lower work function of h-BN:Ge-O indicates the upward shift of the Fermi level of h-BN:Ge-O closer to the CBM, demonstrating the n-type nature.” The conclusion about the n-type nature is too strong considering the results of the UPS measurements and therefore this statement should be revised.

Authors reply:

We thank the Reviewer point out this improper statement. We have revised the statement accordingly.

Actions taken:

We have revised:

Line 9, Page 12, “The lower work function of h-BN:Ge-O indicates the upward shift of the Fermi level of h-BN:Ge-O closer to the CBM, demonstrating the n-type nature.” → “The lower work function of h-BN:Ge-O indicates the upward shift of the Fermi level of h-BN:Ge-O closer to the CBM, indicating the transition from intrinsic h-BN into *n*-type conduction.”

- **Reviewer #1 wrote:** 3. The authors do not comment on the Ge-O doping level, while it is well known that this is a key parameter in the optimal doping. The other question related to this issue is the character of conductivity. Is it hopping conductivity or regular electron conductivity in the conduction band?

Authors reply:

This is a good suggestion. According to the Reviewer's suggestion, we tried to determine the doping level of Ge-O in our samples. Due to the monolayer thickness of h-BN, accurate composition measurement such as SIMS meets its difficulty in obtain a reliable doping concentration in h-BN. This is a common challenge for 2D materials. Therefore, we employed Auger electron spectroscopy (AES) and XPS to determine the Ge-O doping level, which are both regarded as the powerful surface-analytical techniques. Based on the recorded AES data (**Figure 3g**), the doping level of Ge for B is calculated to be about 4.1%. From the XPS result (**Figure 3h-k**), the Ge doping level is determined to be about 2.8%. They indicate a high doping concentration of Ge-O. We have added this result into the revised manuscript.

As for the conduction type, our discussion is as follows. For nitride semiconductors, regular band conduction should be dominant at room temperature whereas the hopping conduction is dominant at low temperature [*Journal of Applied Physics* 87, 1832 (2000)]. In h-BN:Ge-O system, the shallowed Ge-O donor level has lower activation energy, which will strongly prefer the band conduction. On the other hand, due to the heavy doping of Ge-O in h-BN (>2.8%), the conduction mechanism could also involve partially hopping conductivity at room temperature [*Japanese Journal of Applied Physics* 58, 098004 (2019)] Overall, the dominance of band conduction should still hold. We have added discussions in the revised manuscript.

Actions taken:

We have added:

- (1) **Line 5, Page 10**, "Since the doping level is crucial parameter for the *n*-type h-BN, XPS measurement was employed to estimate the Ge-O doping concentration. The substitutional doping level of Ge_B is about 2.8%, which indicates a heavy doping and is consistent with the absorption spectrum (Fig. 3e)."
- (2) **Line 23, Page 11**, "In addition, due to the shallowed Ge-O donor level has lower activation energy, regular band conductivity should be dominant for h-BN:Ge-O at room temperature. However, due to the heavy doping of Ge-O (>2.8%), the conduction mechanism could be partially contributed by hopping conductivity.^{45,46}"
- (3) **References:**
[*Japanese Journal of Applied Physics* 58, 098004 (2019)] as new reference [45].
[*Journal of Applied Physics* 87, 1832 (2000)] as new reference [46].

- **Reviewer #1 wrote:** 4. What about Ge and O concentration in the reference samples?

Authors reply:

We further performed the AES and XPS measurements on the undoped h-BN films, as shown in **Figure R2**. The Ge signal cannot be detected from both the AES and XPS spectra, indicating that the absence of Ge in the reference h-BN samples. The O signal usually involves the signal from the O contamination, which is not reliable for comparison.

Figure R2. (a) AES spectrum of the undoped h-BN film. (b)-(c) XPS spectra of the h-BN:Ge-O and undoped h-BN films, showing the Ge 3d level (b) and Ge 2p level (c).

Actions taken:

We have added:

- (1) **Line 8, Page 10**, “By comparison, the absence of Ge in the undoped h-BN was confirmed (Fig. S20).”
- (2) **Supplementary Information**, Figure R2 → Figure S20.

- **Reviewer #1 wrote:** 5. The strong absorption at 250 nm, see Fig.3 (e) and S15, seems to be very important and is too shortly commented in the paper. What is the origin of this absorption? Which impurity level is responsible for this absorption? What about acceptor-like states in these samples and their compensation by donor-like states?

Authors reply:

It is right that the detailed discussions on this important absorption peak is necessary. The strong absorption at 250 nm is most likely related with the Ge-O energy levels. In order to confirm the origin of this absorption, we further measured absorption spectra of h-BN:O, h-BN:Ge, and undoped h-BN samples for comparison, as shown in **Figure R3**. It can be seen that the sharp band-edge absorption peaks of intrinsic h-BN is at 209.6 nm (5.93 eV). The impurity related absorption peaks for h-BN:Ge-O, h-BN:Ge and h-BN:O appear at 5.53 eV, 5.19 eV and 4.89 eV, respectively. These results distinguish the different origins. Detailed examination further shows that the absorption shoulder of h-BN:Ge-O is abroad, covering the near band-edge band down to the h-BN:O related peak. This may mean that in the h-BN:Ge-O sample, the absorptions may include transitions from Ge, O, and Ge-O impurity levels. Of course, the Ge-O type impurity dominates.

Figure R3. Absorption spectra of h-BN:O, h-BN:Ge, h-BN:Ge-O, and undoped h-BN films.

Figure R4. Energy band diagram showing the transitions responsible for the absorption peaks of doped h-BN.

The absorptions should stem from the transitions between the dopant levels (Ge-O, Ge, O) and the acceptor-like/valence band levels, as summarized in **Figure R4**. It was reported that the boron vacancy (V_B) could be an acceptor contributing to the p-type conduction [*Nano Lett.*, 17, 3738–3743 (2017)]. Therefore, the possible acceptor-like state could be the boron vacancy level. However, pertinent assignment of transition levels involved in the absorption requires further analysis based on fine structures of low-temperature PL spectra. This issue will be studied as a following work.

Actions taken:

We have revised:

- (1) **Line 2, Page 9**, “which should be attributed to the impurity level related absorption.” → “which should be attributed to the Ge-O impurity level related absorption (see Fig. S16 and related discussion).”

We have added:

- (2) **Supplementary Information**: Figure R3-4 → Figure S16.

Discussions on this issue were added accordingly, “In order to confirm the origin of the absorption in h-BN:Ge-O, we further measured absorption spectra of h-BN:O, h-BN:Ge, and undoped h-BN samples for comparison...”

- **Reviewer #1 wrote**: 6. n-hBN/p-GaN structure was studied, but details on p-type layer (level of Mg doping etc.) are not provided. Moreover, this structure and measurements on this structure are not commented on in the context of previous research for h-BN/GaN structures, see ACS Applied Materials & Interfaces 2022, 14, 4, 6131-613.

Authors reply:

According to the Reviewer’s helpful suggestion, we further performed SIMS test for the *p*-GaN epilayer to determine the doping level of Mg, as shown in **Figure R5**. The *p*-type GaN layer was grown above an AlGaN basal layer and the thickness of *p*-GaN is about 20 nm. The Mg doping concentration in the *p*-GaN is at the level of 10^{19} cm⁻³. The resistivity of the *p*-layer is about 6.27 Ω cm based on the electrical characterization.

We carefully read the literature mentioned by the Reviewer and cited with comments in the revised manuscript. The Schottky diodes based on undoped h-BN/GaN were reported in this paper.

Figure R5. (a) Element concentration and (b) element composition of the *p*-GaN/AlGaN epilayer.

Actions taken:

We have added:

- (1) **Epitaxial growth, Page 17**, “The *p*-GaN was grown above an AlGaN template and the thickness of *p*-GaN is about 20 nm. The Mg doping concentration is at the level of 10^{19} cm⁻³ and the resistivity is about 6.27 Ω cm in the *p*-GaN (Fig. S25). The size of the device is 400×400 μm.”
- (2) **Supplementary Information**: Figure R5 → Figure S25.
- (3) **Line 3, Page 13**, “In previous work, the Schottky diodes based on undoped hBN/GaN were reported.⁵³”

(4) **References:**

[ACS Appl. Mater. Interfaces, 14, 6131–6137 (2022)] as new reference [53].

- **Reviewer #1 wrote:** 7. Is there a “driving force” other than statistics that causes Ge to bind to O? This issue should be better commented on in this article.

Authors reply:

According to the Reviewer’s helpful suggestion, we further performed first-principle calculations on the formation energies of individual Ge_B impurity, $\text{Ge}_\text{B}\text{-O}_\text{N}$ dimer, and $\text{Ge}_\text{B}\text{-2O}_\text{N}$ trimer in h-BN monolayer to investigate the driving force of O binding. The formation energies of impurity X is defined as follow [J. Appl. Phys., 95, 3851–79 (2004)]:

$$E^f[X] = E_{\text{tot}}[X] - E_{\text{tot}}[\text{original}] - \sum_i n_i \mu_i,$$

where $E_{\text{tot}}[X]$ is the total energy derived from the system with impurity X doping, $E_{\text{tot}}[\text{original}]$ is the total energy of the system before the impurity X doping, n_i indicates the number of atoms of type i that have been added ($n_i > 0$) or removed ($n_i < 0$) from the supercell when the impurity is formed, and the μ_i are the chemical potentials of these species.

Figure R6. Formation energies of Ge_B , $\text{Ge}_\text{B}\text{-O}_\text{N}$, $\text{Ge}_\text{B}\text{-2O}_\text{N}$, and $\text{Ge}_\text{B}\text{-3O}_\text{N}$ in h-BN.

The simulation results are shown in **Figure R6**. It is interesting to see that the O binding will lower the formation energy of Ge or Ge-O impurities. The decreasing formation energy follows the trend: $\text{Ge}_\text{B} > \text{Ge}_\text{B}\text{-O}_\text{N} > \text{Ge}_\text{B}\text{-2O}_\text{N} > \text{Ge}_\text{B}\text{-3O}_\text{N}$. This implies that the binding and coupling with O could minimized the formation energy of Ge, which is the driving force for the formation preference of Ge-O bonds. More importantly, the formation energies of single O binding to $\text{Ge}_\text{B}\text{-O}_\text{N}$ dimer, and O binding to $\text{Ge}_\text{B}\text{-2O}_\text{N}$ trimer are reduced to -1.06 eV and -0.98 eV, respectively. The negative formation energies indicate that the Ge-O_2 and Ge-O_3 are most likely to form during the growth. These simulation results strongly prove that there is a driving force that prompts the O binding to the Ge donor. Therefore, not only for the statistics, the Ge-O_2 trimer and Ge-O_3 tetramer doping are indeed favorable in the thermodynamics aspect. On the other hand, in the experiments, Ge_2O_3 was employed as the precursor for Ge-O doping, which actually already have the Ge-O binding before incorporation into the h-BN.

We have added discussions in the revised manuscript as well as the Supplementary Information.

Actions taken:

We have added:

(1) **Line 12, Page 10**, “Moreover, based on the first-principles calculations, the decreasing formation energy follows the trend of $\text{Ge}_B > \text{Ge}_B\text{-O}_N > \text{Ge}_B\text{-2O}_N > \text{Ge}_B\text{-3O}_N$. Therefore, the Ge-O_2 trimer and Ge-O_3 tetramer are also energetically favorable in the thermodynamics aspect (see Fig. S21 and related discussion).”

(2) **Supplementary Information:**

Figure R6 → Figure S21

Discussions on this issue were added accordingly, “We further performed first-principle calculations on the formation energies of individual Ge_B impurity, $\text{Ge}_B\text{-O}_N$ dimer, and $\text{Ge}_B\text{-2O}_N$ trimer in h-BN monolayer to investigate the driving force of O binding...”

- **Reviewer #1 wrote:** 8. Why are Hall's measurements not made on these samples? Are there problems with ohmic contacts to h-BN? This issue should be commented on in this article.

Authors reply:

The Reviewer is right that Hall measurement is important. However, ohmic contact of electrodes is not well obtained, as shown in the I-V curve of **Figure 4e**. This can be contributed to Au electrode we used in the I-V test. The work function of Au is about 5.1 eV whereas the work function of *n*-type h-BN is about 4.10 eV.

Instead of Hall measurement, we obtained further electrical properties of the *n*-type h-BN monolayer by the FET device method [*ACS Nano* 5, 5, 3591–3598 (2011)]. The electron mobility of *n*-type h-BN monolayer is calculated to be $0.014 \text{ cm}^2 \text{ V}^{-1} \text{ s}^{-1}$. The resistivity (ρ) of *n*-type h-BN monolayer is about $2.29 \times 10^4 \text{ } \Omega \text{ cm}$ and the electron concentration can reach $1.94 \times 10^{16} \text{ cm}^{-3}$.

We have added above discussions to the revised manuscript.

Actions taken:

We have added:

- (1) **Line 11, Page 12**, “Ohmic contact of electrodes is not well obtained due to the work function difference between Au electrode (5.1 eV) and *n*-type h-BN (4.10 eV), which make it difficult for Hall measurements.”
- (2) **Line 18, Page 12**, “Electrical properties of the *n*-type h-BN monolayer was obtained by FET device method⁵² (see details in **Supplementary Information**). The electron mobility and concentration are determined to be $0.014 \text{ cm}^2 \text{ V}^{-1} \text{ s}^{-1}$ and $1.94 \times 10^{16} \text{ cm}^{-3}$, respectively.”
- (3) **References:**
[*ACS Nano* 5, 5, 3591–3598 (2011)] as new references [52].
- (4) **Supplementary Information:**

Discussions on this issue were added to the **Supplementary method about the FET device** accordingly, “Instead of Hall measurement, we obtained further electrical properties of the *n*-type h-BN monolayer by the FET device method...”

- **Reviewer #2 wrote:** This work reports on a study aiming at demonstrating the n-type doping of h-BN epitaxial films. The latter are of importance and are the subject of strong interest due to their potential for various applications, such as 2D heterostructures and deep UV LEDs and lasers. Efficient doping of wide bandgap materials is challenging. This is especially the case of h-BN for which it is not completely clear that both p and n doping have been achieved despite numerous (and contradictory) theoretical and experimental works. The paper provides an impressive and almost complete bundle of theoretical and experimental results contributing to the demonstration of n-doped h-BN. The approach followed by the authors is very convincing and significant. Using first-principles calculations, the authors first investigated the strong orbital coupling between donor impurity atom and various coordinating atoms leading to the identification of Ge-O dimer, trimer and tetramer as efficient donors. They then grew the corresponding epitaxial films using LP-CVD technique. HRTEM, SAED, Raman, AES, XPS were then used to confirm the chemical doping. Finally, electrical measurements on h-BN films and p-GaN/n-BN were carried out to assess the electrical doping. Such a deep and rather complete approach is completely original for the study of n-doping in h-BN although similar approach has been used for p-doping.

Authors reply:

We sincerely thank the Reviewer for his/her careful reading and high appreciation of the value of our work, providing helpful suggestions, and especially the kind recommendation for the publication of our manuscript. We have considered all the comments and suggestions and revised the manuscript accordingly.

- **Reviewer #2 wrote:** There is no doubt that the chemical doping has been achieved and that the conductivity has been altered by the Ge-Ox incorporation in the h-BN lattice. The evidences are clear. Nevertheless, Hall measurements would have been significant and very useful to estimate both the doping concentration and carrier mobility. Also, since p-BN is achievable why have the authors chosen the fabrication of a n-BN/p-GaN heterostructure instead of a p-BN/n-BN homojunction? In addition, some points require more clarification.

Authors reply:

The Reviewer is right that Hall measurement is important. However, the ohmic contact of electrodes is not well obtained, as shown in the I-V curve of **Figure 4e**. This can be contributed to Au electrode we used in the I-V test. The work function of Au is about 5.1 eV whereas the work function of n-type h-BN is about 4.10 eV. Alternatively, we obtained further electrical properties of the n-type h-BN monolayer by the FET device method [ACS Nano 5, 5, 3591–3598 (2011)]. According to the FET results, the electron mobility and electron concentration were determined to be $0.014 \text{ cm}^2 \text{ V}^{-1} \text{ s}^{-1}$ and $1.94 \times 10^{16} \text{ cm}^{-3}$, respectively. This result has been added to the revised manuscript.

As the Reviewer mentioned, p-hBN/n-hBN homojunction should be fabricated. Eventually, we have done the fabrication. However, several unusual phenomena have been observed from this homojunction, which still could not be well explained by available data. The study on p-hBN/n-hBN homojunction is still undergoing with a double-check and extended characterizations. Therefore, we first show the results of the n-BN/p-GaN heterostructure instead of a p-BN/n-BN homojunction.

Actions taken:

We have added:

- (1) **Line 11, Page 12**, “Ohmic contact of electrodes is not well obtained due to the work function difference between Au electrode (5.1 eV) and *n*-type h-BN (4.10 eV), which make it difficult for Hall measurements.”
 - (2) **Line 18, Page 12**, “Electrical properties of the *n*-type h-BN monolayer was obtained by FET device method⁵² (see details in **Supplementary Information**). The electron mobility and concentration are determined to be 0.014 cm² V⁻¹ s⁻¹ and 1.94×10¹⁶ cm⁻³, respectively.”
 - (3) **References:**
[ACS Nano 5, 5, 3591–3598 (2011)] as new references [52].
 - (4) **Supplementary Information:**
 - (5) Discussions on this issue were added to the **Supplementary method about the FET device** accordingly, “Instead of Hall measurement, we obtained further electrical properties of the *n*-type h-BN monolayer by the FET device method...”
- **Reviewer #2 wrote:** C-V measurements: Fig.5-f shows the C-V for different AC frequencies. It is reported an increase of the capacitance with the frequency. This is unexpected. The capacitance should not depend on frequency except in presence of defect which would tend to decrease it. What is the origin of such a phenomenon? At large reverse voltage, no saturation of the capacitance is observed. Does it mean that the *n*-doping is weak? The 1/C2=f(V) plot would have been useful to confirm the voltage threshold of the diode.

Authors reply:

This question as well as the suggestion on 1/C2=f(V) plot is very helpful to us. According to the Reviewer’s inspiration, we carefully reconsidered this issue and tried to figure out the underlying origin. The phenomenon of increasing capacitance with the increasing frequency is unusual for a *p-n* junction indeed. Generally, the capacitance should be constant over frequency. From literature study, we are happy to find that when the applied frequencies approach the capacitor’s self-resonant frequency, a parasitic series inductance will work and result in an effective capacitance (C_E) that is larger than the nominal capacitance (C_O) [Measurement Techniques, 15, 1046–1049 (1972)] [Measurement Techniques, 6, 758–761, 1963] [An Analysis of Precision Methods of Capacitance Measurements at High Frequency, Creative Media Partners, LLC, 2021].

Figure R7. The lumped element equivalent model. C_o is the nominal capacitance, L_c is the parasitic series inductance, and R_t is equivalent resistance.

The lumped element equivalent model is illustrated in **Figure R7** and the effective capacitance can be described as:

$$C_E = \frac{C_O}{1 - (2\pi f)^2 L_C C_O}, \quad (R1)$$

where f is the applied frequency, L_C is the parasitic series inductance of the capacitor. In practice the inductance of the capacitors is rather small so that $C_E = C_O$. When the inductance becomes considerable, at high frequencies the effective capacitance will increase along with the frequency. Thus, we can find out that the origin of the capacitance increasing in this n -hBN/ p -GaN junction should be attributed to the considerable inner series inductance. By fitting the C_p - f data with eq.(R1), as shown in **Figure R8**, we can obtain the value of the inner series inductance. As a result, the nominal capacitance C_O is 0.2486 pF and the inductance L_C is 0.0035 H. This considerable inner inductance in the n -hBN/ p -GaN junction should be highly related with the unique 2D layered structure of the multilayer h-BN. Because of the weak van der Waals interaction between n -h-BN layers, the possible formation of multiple conducting micro-channels could be the reason to introduce this inner inductance. However, reliable evidences and explanations need further thorough investigations.

Figure R8. (a) C-V curves of the n -hBN/ p -GaN junction under various ac frequencies. (b) Effective capacitance as a function of frequency and the fitting curve.

According to the Reviewer's suggestion, we have plotted the $1/C^2$ -V curve, as shown in **Figure R9**. The $1/C^2$ -V curve appears nonlinear at zero voltage. Therefore, using the extrapolation of the $1/C^2$ -V curve to determine the threshold voltage could be very inaccurate. In fact, the threshold voltage estimated by the extrapolation is as large as 15 V. The nonlinear feature of $1/C^2$ -V curve can be attributed to the inhomogeneous doping profile in the interfacial region of n -hBN/ p -GaN (**Figure R5**). Similar phenomenon has been also observed in non-conventional p - n junctions such as ZnO nanorod/ p -Si diode [*Elektronika Ir Elektrotehnika*, 27(4), 49-54 (2021)] [*Semicond. Sci. Technol.*, 28, 125006 (2013)].

Figure R9. C-V curve measured from reverse to forward bias of the n -hBN/ p -GaN junction under a frequency of 3 MHz and corresponding $1/C^2$ plot versus voltage.

Actions taken:

We have added:

- (1) **Line 22, Page 14**, “The phenomenon of increasing capacitance with the increasing frequency is unusual for a p-n junction. This may stem from the existence of a considerable inner series inductance⁵⁹ within n-h-BN layers, which is determined to be about 0.0035 H (See details in Supplementary Information Fig. S27).”
- (2) **Line 2, Page 15**, “In addition, the $1/C^2$ -V curve is shown in Fig. 5f, which appears nonlinear at zero voltage. The nonlinear feature of $1/C^2$ -V curve can be attributed to the inhomogeneous doping profile in the interfacial region of n-hBN/p-GaN.”
- (3) Figure R9 → Fig. 5f
- (4) **References**:
[*Measurement Techniques*, 6, 758–761, 1963] as new reference [59].
- (5) **Supplementary Information**, Figure R8b → Figure S27.

Detailed discussions on the effective capacitance under various frequencies were provided, “The phenomenon of increasing capacitance with the increasing frequency is unusual for a p-n junction...”

- **Reviewer #2 wrote:** I-V measurements: The size of the devices is not indicated. The doping of p-GaN is not given. It is probably much larger than that of n-h-BN. Thus, the n-h-BN layer should be completely depleted (for sure for the single layer and probably for the 6-multi-layers also). How does it impact the diode behavior?

Authors reply:

According to the Reviewer’s recommendation, additional parameters of the devices were provided. The size of the device is $400 \times 400 \mu\text{m}$. The thickness, Mg doping concentration, and resistivity of p-GaN are 20 nm, $\sim 10^{19} \text{ cm}^{-3}$, and $6.27 \Omega \text{ cm}$, respectively. This information has been added to the revised manuscript and listed in the change list.

Figure R10. Energy band diagram, space charge distribution, and electron field distribution of p-GaN/n-hBN junction in thermal equilibrium with (a) thick h-BN layer and (b) ultrathin h-BN layer.

We strongly agree with the Reviewer's opinion on the depletion region in p - n junction, at least traditionally. The doping level and carrier density of p -GaN should be larger than those of n -type h-BN layer. The energy band diagram, space charge distribution, and electric field distribution in this p -GaN/ n -hBN junction in thermal equilibrium are shown in **Figure R10**. For a thick n -hBN layer, the n -hBN layer is not completely depleted (a) whereas the ultrathin n -hBN should be largely depleted (b) for sure, as mentioned by the Reviewer. From the electric field distribution in **Figure R10**, we can see that the built-in electric field E_m at the depleted ultrathin h-BN interface will decrease but still work for the typical diode behavior. This decreasing built-in field with the decreasing h-BN thickness could lead to decreasing the forward turn-on voltage, which has been confirmed by the I-V test, as shown in **Figure 5(d)-(e)**. The forward turn-on voltage for the monolayer h-BN case (1.44 eV) is smaller than that the 6-layer n -h-BN case (3.11 eV).

Actions taken:

We have added:

- (1) **Epitaxial growth, Page 17**, “The p -GaN was grown above an AlGaN template and the thickness of p -GaN is about 20 nm. The Mg doping concentration is at the level of 10^{19} cm⁻³ and the resistivity is about 6.27 Ω cm in the p -GaN (Fig. S25). The size of the device is 400×400 μ m.”
- (2) **Line 2, Page 14**, “The underlying mechanism is rooted in the depletion region and the built-in electric-field. For the 1-monolayer n -hBN device, such an ultrathin n -type h-BN layer should be completely depleted and built-in field will decrease with the decreasing h-BN thickness (see Fig. S26 and related discussion). Thus, the forward turn-on voltage for the monolayer h-BN case is smaller than that the 6-layer n -h-BN case.”
- (3) **Supplementary Information: Figure R5** → **Figure S25**.
- (4) **Supplementary Information: Figure R10** → **Figure S26**.
Related discussions were put in the context, “The doping level and carrier density of p -GaN should be larger than those of n -type h-BN layer. This may impact the diode behavior...”

- **Referee #2 wrote:** h-BN monolayer: Its thickness is said to be 0.59 nm. This value is twice the one found in the literature. Why?

Authors reply:

In principle, the ideal thickness of monolayer h-BN is about 0.31 nm. However, the sample in ambient would have moist layer on the surface which will slight increase the measured thickness by AFM. This phenomenon had been commonly observed for single-layer h-BN, graphene and other 2D materials [*Nature* 579, 219–223 (2020)] [*ACS Nano*, 11, 12057–12066 (2017)]. In some reports, the AFM recorded thickness for monolayer h-BN even can reach ~ 1 nm [*Nano Lett.*, 14, 839–846 (2014)]. Usually, an AFM thickness below 1 nm could correspond to a monolayer thickness.

- **Referee #2 wrote:** p -doping of h-BN: It is said that it can be easily achieved. It seems that it is not the case. Some controversy still exists, and the p -doping is not very efficient.

Authors reply:

We agree with the Reviewer's opinion. Some literatures have shown the inefficient p -type doping using Mg [*Nanomaterials*, 11, 211 (2021)]. According the Reviewer's comment, we have revised the statement.

Actions taken:

We have revised:

- (1) **Line 24, Page 2**, “The *p*-type h-BN can be easily achieved by either Mg-doping or producing boron vacancies.” → “The *p*-type h-BN has been achieved by either Mg-doping or producing boron vacancies.”

- **Referee #2 wrote:** Others: Page 8, line 1: Could you define what an antioxidant test is? Fig.4-c: It is said that the conductivity is uniform, but it does not look like that. Last, the title of the paper is a little weird and the English language requires improvements.

Authors reply:

Because h-BN has high resistance to oxidation, it can be used to protect Cu foil from oxidation in ambient. During the antioxidant test, the bare Cu foil and Cu foil covered with fully coalesced h-BN monolayer were placed on a heating plate, then the samples were heated in air at 200 °C for 10 minutes. After the heating, the bare Cu foil shows serious oxidization by the air, exhibiting a significant color change from yellow to red. In contrast, the h-BN protected Cu foil remains unoxidized with original color. We have added experimental details in the Methods section.

The Reviewer is right that As for Figure 4c, it is inappropriate to describe “uniform”. We have revised that description.

According to the Reviewer’s kind suggestion, we have revised the title as “**Towards *n*-Type Conductivity in Hexagonal Boron Nitride**” and asked an English native speaker to proofread our manuscript with revisions. Revisions are also summarized below in the List of Changes and marked in the revised manuscript.

Actions taken:

We have added:

- (1) **Methods section, Page 17**, “For the antioxidant test, the bare Cu foil and Cu foil covered with fully coalesced h-BN monolayer were placed on a heating plate, then the samples were heated in air at 200 °C for 10 minutes.”

We revised:

- (2) **Title:** “Towards *n*-Type Conductivity in Hexagonal Boron Nitride”
- (3) **Line 9, Page 11**, “Meanwhile, this conduction has uniformly covered the whole area of h-BN film...” → “Meanwhile, this conduction has covered the whole area of h-BN film...”.

List of Changes

We have revised:

- (1) **Title:** “Towards n-Type Conductivity in Hexagonal Boron Nitride”
- (2) **Line 23, Page 2,** “The *p*-type h-BN can be easily achieved by either Mg-doping or producing boron vacancies.” → “The *p*-type h-BN has been achieved by either Mg-doping or producing boron vacancies.”
- (3) **Line 14, Page 5,** “According to the principle of total energy conservation, the energy of the π^* level could be pushed up if at the same time the π level is pulled downward.” → “According to the principle of total energy conservation as well as the level repulsion between different orbitals,^{19,30} if the π level is pulled downward the energy of the π^* level could be pushed up.”
- (4) **Line 2, Page 9,** “which should be attributed to the impurity level related absorption.” → “which should be attributed to the Ge-O impurity level related absorption (see Fig. S16 and related discussion).”
- (5) **Line 9, Page 11,** “Meanwhile, this conduction has uniformly covered the whole area of h-BN film...” → “Meanwhile, this conduction has covered the whole area of h-BN film...”
- (6) **Line 9, Page 12,** “The lower work function of h-BN:Ge-O indicates the upward shift of the Fermi level of h-BN:Ge-O closer to the CBM, demonstrating the n-type nature.” → “The lower work function of h-BN:Ge-O indicates the upward shift of the Fermi level of h-BN:Ge-O closer to the CBM, indicating the transition from intrinsic h-BN into *n*-type conduction.”
- (7) **Line 19, Page 23,** “This work was partly supported by the National Key Research and Development Program of China (2016YFB0400800), the National Natural Science Foundation of China (61574116, 61974124, 61675172 and 11804115), and the Natural Science Foundation of Fujian Province (2017J01124) of China.” → “This work was partly supported by the National Key Research and Development Program (2016YFB0400800), the National Natural Science Foundation (62074133, 61974124 and 11804115), and the Science and Technology Programs of Fujian Province (2021H0001 and 2017J01124) of China.”

We have added:

- (1) **References**
[*Physical Review Letters*, 98, 135506 (2007)] as reference [19].
[*Applied Physics Express*, 6, 042104 (2013)] as new reference [30].
[*Japanese Journal of Applied Physics* 58, 098004 (2019)] as new reference [45,46].
[*ACS Nano* 5, 5, 3591–3598 (2011)] as new references [52].
[*ACS Appl. Mater. Interfaces*, 14, 6131–6137 (2022)] as new reference [53].
[*Measurement Techniques*, 6, 758–761, 1963] as new reference [59].

We have added:

- (1) **Line 5, Page 10,** “Since the doping level is crucial parameter for the *n*-type h-BN, XPS measurement was employed to estimate the Ge-O doping concentration. The substitutional doping level of Ge_B is about 2.8%, which indicates a heavy doping and is consistent with the absorption spectrum (Fig. 3e).”
- (2) **Line 8, Page 10,** “By comparison, the absence of Ge in the undoped h-BN was confirmed (Fig. S20).”
- (3) **Line 12, Page 10,** “Moreover, based on the first-principles calculations, the decreasing

formation energy follows the trend of $\text{Ge}_B > \text{Ge}_B\text{-O}_N > \text{Ge}_B\text{-2O}_N > \text{Ge}_B\text{-3O}_N$. Therefore, the Ge-O_2 trimer and Ge-O_3 tetramer are also energetically favorable in the thermodynamics aspect (see Fig. S21 and related discussion).”

- (4) **Line 23, Page 11**, “In addition, due to the shallowed Ge-O donor level has lower activation energy, regular band conductivity should be dominant for h-BN:Ge-O at room temperature. However, due to the heavy doping of Ge-O (>2.8%), the conduction mechanism could be partially contributed by hopping conductivity.^{45,46}”
- (5) **Line 11, Page 12**, “Ohmic contact of electrodes is not well obtained due to the work function difference between Au electrode (5.1 eV) and *n*-type h-BN (4.10 eV), which make it difficult for Hall measurements.”
- (6) **Line 18, Page 12**, “Electrical properties of the *n*-type h-BN monolayer was obtained by FET device method⁵² (see details in **Supplementary Information**). The electron mobility and concentration are determined to be $0.014 \text{ cm}^2 \text{ V}^{-1} \text{ s}^{-1}$ and $1.94 \times 10^{16} \text{ cm}^{-3}$, respectively.”
- (7) **Line 3, Page 13**, “In previous work, the Schottky diodes based on undoped hBN/GaN were reported.⁵³”
- (8) **Line 2, Page 14**, “The underlying mechanism is rooted in the depletion region and the built-in electric-field. For the 1-monolayer *n*-hBN device, such an ultrathin *n*-type h-BN layer should be completely depleted and built-in field will decrease with the decreasing h-BN thickness (see Fig. S26 and related discussion). Thus, the forward turn-on voltage for the monolayer h-BN case is smaller than that the 6-layer *n*-h-BN case.”
- (9) **Line 22, Page 14**, “The phenomenon of increasing capacitance with the increasing frequency is unusual for a p-n junction. This may stem from the existence of a considerable inner series inductance⁵⁹ within *n*-h-BN layers, which is determined to be about 0.0035 H (See details in Supplementary Information Fig. S27).”
- (10) **Line 2, Page 15**, “In addition, the $1/C^2$ -V curve is shown in Fig. 5f, which appears nonlinear at zero voltage. The nonlinear feature of $1/C^2$ -V curve can be attributed to the inhomogeneous doping profile in the interfacial region of *n*-hBN/*p*-GaN.”
- (11) **Epitaxial growth, Page 17**, “The *p*-GaN was grown above an AlGaN template and the thickness of *p*-GaN is about 20 nm. The Mg doping concentration is at the level of 10^{19} cm^{-3} and the resistivity is about $6.27 \text{ } \Omega \text{ cm}$ in the *p*-GaN (Fig. S25). The size of the device is $400 \times 400 \text{ } \mu\text{m}$.”
- (12) **Methods section, Page 17**, “For the antioxidant test, the bare Cu foil and Cu foil covered with fully coalesced h-BN monolayer were placed on a heating plate, then the samples were heated in air at $200 \text{ }^\circ\text{C}$ for 10 minutes.”
- (13) **Line 20, Page 24**, “Formation energies of impurities in h-BN; SIMS results of *p*-GaN; Band diagram, space charge and electron field distributions of *p*-GaN/*n*-h-BN junction; Effective capacitance of the junction; Supplementary method about the FET device”
- (14) **Figure R9** → Fig. 5f, “and corresponding $1/C^2$ plot versus voltage under a frequency of 3 MHz.”

We have added:

- (1) **Supplementary Information**, Figure R2 → Figure S20.
- (2) **Supplementary Information**: Figure R3-4 → Figure S16.

Discussions on this issue were added accordingly, “In order to confirm the origin of the

absorption in h-BN:Ge-O, we further measured absorption spectra of h-BN:O, h-BN:Ge, and undoped h-BN samples for comparison...”

(3) **Supplementary Information:** Figure R5 → Figure S25.

(4) **Supplementary Information:**

Figure R6 → Figure S21

Discussions on this issue were added accordingly, “We further performed first-principle calculations on the formation energies of individual Ge_B impurity, Ge_B-O_N dimer, and Ge_B-2O_N trimer in h-BN monolayer to investigate the driving force of O binding...”

(5) **Supplementary Information,** Figure R8b → Figure S27.

Detailed discussions on the effective capacitance under various frequencies were provided, “The phenomenon of increasing capacitance with the increasing frequency is unusual for a *p-n* junction...”

(6) **Supplementary Information:** Figure R10 → Figure S26.

Related discussions were put in the context, “The doping level and carrier density of *p*-GaN should be larger than those of *n*-type h-BN layer. This may impact the diode behavior...”

(7) **Supplementary Information:**

Discussions on this issue were added to the **Supplementary method about the FET device** accordingly, “Instead of Hall measurement, we obtained further electrical properties of the *n*-type h-BN monolayer by the FET device method...”

REVIEWERS' COMMENTS

Reviewer #1 (Remarks to the Author):

The authors revised the manuscript according to my comments and suggestions. The replies and changes made to the manuscript are satisfactory and I have no further comments. I can recommend this paper for publication in Nature Communications as is.

Reviewer #2 (Remarks to the Author):

All my concerns have been very well addressed in the revisions. Thanks to the authors for this effort.

I just have a last question relating to a comment of reviewer #1. In the corresponding answer it is mentioned that "the substitutional doping level of Ge(B) is about 2.8%". Is the material still a h-BN compound or rather a new alloy h-B(GeO)N?

Response Letter

We sincerely thank the Reviewers for helpful suggestions, high appreciation of the value and also the kind recommendation of the publication of our work. We have carefully considered the Reviewer's further comment, answered the question and revised manuscript accordingly.

Reply to the report of **Reviewer #1** on NCOMMS-22-05830A/Lu et al.

- **Reviewer #1 wrote:** The authors revised the manuscript according to my comments and suggestions. The replies and changes made to the manuscript are satisfactory and I have no further comments. I can recommend this paper for publication in Nature Communications as is.

Authors reply:

We sincerely thank the Reviewer for his/her kind recommendation for the publication of our paper.

Reply to the report of **Reviewer #2** on NCOMMS-22-05830A/Lu et al.

- **Reviewer #1 wrote:** All my concerns have been very well addressed in the revisions. Thanks to the authors for this effort. I just have a last question relating to a comment of reviewer #1. In the corresponding answer it is mentioned that "the substitutional doping level of Ge(B) is about 2.8%". Is the material still a h-BN compound or rather a new alloy h-B(GeO)N?

Authors reply:

We sincerely thank the Reviewer for his/her careful consideration and deep thinking on the issues in our manuscript. The comment mentioned by the Reviewer is very important. Here we demonstrate the absorption spectra of the Ge-O doped h-BN together with that of the undoped h-BN [**Figure R1(c)**], which strongly confirm that the h-BN:Ge-O is still a h-BN compound with Ge-O doping instead of a new alloy.

Firstly, the doping concentration could be overestimated by AES or XPS estimation due to the ultrathin thickness of h-BN monolayer and the weak signal acquired by these two surface characterization techniques.

Secondly, if the Ge-O doped h-BN is a new alloy, in principles, the optical properties should be dramatically changed. For example, Ram *et al.* [*Appl. Phys. Lett.* 104, 163101 (2014)] have reported the band gap narrowing effect of h-BN samples exposed to O₂-plasma, as shown in **Figure R1(a)**. Weng *et al.* [*Adv. Mater.* 2017, 29, 1700695] also reported that the band gap of h-BN is narrowed from 5.5 eV down to only 2.1 eV when a new BNO alloy forms, as shown in **Figure R1(b)**. The absorption spectra of h-BN:Ge-O and undoped h-BN are shown in **Figure R1(c)**. We can clearly see that the band edge absorption of h-BN at ~ 5.93 eV is still dominant for the h-BN:Ge-O sample and is consistent with that detected in the undoped h-BN sample. The Ge-O doping introduces an impurity peak at the lower energy side to the band

edge peak, which strongly indicates the role of Ge-O as an impurity rather than an alloy.

According to the Reviewer's comment, we have added discussions in:

Line 8, Page 10, "The actual doping concentration could be overestimated by XPS due to the ultrathin thickness of h-BN monolayer. The absorption spectrum of the h-BN:Ge-O showed an impurity peak and a dominant band edge peak at ~ 5.93 eV, consistent with that of the intrinsic h-BN (Fig. 3e). This indicates that the Ge-O are incorporated into h-BN as dopants instead of forming a new alloy."

Figure R1. (a) Absorption spectra acquired from h-BN samples exposed to O_2 -plasma. The inset shows the spectrum of pristine h-BN sample before exposure. [*Appl. Phys. Lett.* 104, 163101 (2014)] (b) Absorption spectra of the synthesized BNO in comparison with standard h-BN. [*Adv. Mater.* 2017, 29, 1700695] (c) Absorption spectra of undoped and Ge-O doped h-BN monolayers in our work.